# Rehabilitation after a Complete Avulsion of the Proximal Rectus Femoris Muscle: Considerations from a Case Report

**DOI:** 10.3390/ijerph18168727

**Published:** 2021-08-18

**Authors:** Christian Baumgart, Casper Grim, Rafael Heiss, Philipp Ehrenstein, Jürgen Freiwald, Matthias Wilhelm Hoppe

**Affiliations:** 1Department of Movement and Training Science, University of Wuppertal, Fuhlrottstraße 10, 42119 Wuppertal, Germany; freiwald@uni-wuppertal.de; 2Department of Orthopedic and Trauma Surgery, Klinikum Osnabrück, Am Finkenhuegel 1, 49076 Osnabrueck, Germany; casper.grim@klinikum-os.de; 3Department of Human Science, University of Osnabrueck, Barbarastrasse 22c, 49076 Osnabrueck, Germany; 4Institute of Radiology, University Hospital Erlangen, Maximiliansplatz 3, 91054 Erlangen, Germany; Rafael.Heiss@uk-erlangen.de; 5Practice OrthoPro Duesseldorf, Breitestr 69, 40213 Duesseldorf, Germany; ehrenstein@orthopro.info; 6Institute of Movement and Training Science, University of Leipzig, Jahnallee 59, 04109 Leipzig, Germany; matthias.hoppe@uni-leipzig.de

**Keywords:** muscle injury, tendon rupture, quadriceps, surgery, strength, electromyography, rehabilitation, isokinetic

## Abstract

Background: A complete avulsion of the proximal rectus femoris muscle is a rare but severity injury. There is a lack of substantial information for its operative treatment and rehabilitation; in particular there is a lack of biomechanical data to evaluate long-term outcomes. Case presentation: The case report presents the injury mechanism and surgical treatment of a complete avulsion of the proximal rectus femoris muscle in a 41-year-old recreational endurance athlete. Moreover, within a one-year follow-up period, different biomechanical tests were performed to get more functional insights into changes in neuromuscular control, structural muscle characteristics, and endurance performance. Within the first month post-surgery, an almost total neuromuscular inhibition of the rectus femoris muscle was present. A stepwise reduction in inter-limb compensations was observable (e.g., in crank torque during cycling) during the rehabilitation. Muscular intra-limb compensations were shown at six months post-surgery and even one year after surgery, which were also represented in the long-term adaption of the muscle characteristics and leg volumes. A changed motor control strategy was shown by asymmetric muscle activation patterns during ergometer cycling, while the power output was almost symmetric. During rehabilitation, there might be a benefit to normalizing neuromuscular muscle activation in ergometer cycling using higher loads. Conclusions: While the endurance performance recovered after six months, asymmetries in neuromuscular control and structural muscle characteristics indicate the long-term presence of inter- and intra-limb compensation strategies.

## 1. Background

The rectus is one portion of the quadriceps femoris muscle. While the origin of the direct head is at the anterior inferior iliac spine, the reflected head originates at the upper region of the acetabulum [1,2]; the insertion is located at the upper pole of the patella [3]. As the only bi-articular muscle of the quadriceps, the rectus femoris contributes to knee extension and hip flexion [4]. The direct head is more involved in the beginning of hip flexion, whereas the reflected head plays a greater role for higher hip flexion angles [1]. For sporting activities, an important morphological point is that the muscle consists of a high percentage of type II fibers [5]. Due to these properties, the rectus femoris muscle is highly stressed during intermittent sports such as soccer, involving sprints, changes of direction, jumps, and ball kicks. Such activities can lead to different severe injuries such as muscle strains or ruptures [4,6]. In particular, high eccentric loading, for example during changes of direction or jumps, can cause severe injuries [3,7]. However, in most previous studies, the exact injury mechanism, which is a crucial basis for the diagnosis, treatment, and subsequent rehabilitation, remains unclear [3].

A complete avulsion of the proximal rectus femoris muscle is rare. The prevalence does not exceed 1% of all quadriceps injuries [6]. Multifactorial risk factors are discussed to predispose for tendon injuries like a previous tear, poor muscle conditioning, inadequate warm-up, muscle fatigue, lifestyle factors, inflammation, genetic factors, vascular impairments, and drugs [1,8]. After a complete avulsion of the proximal rectus femoris, no consensus exists regarding the type of treatment [3,4,9,10]. While a successful return to sport has even been reported after conservative treatment [11,12,13], most patients undergo a surgical repair [3,4,6], whereby the complete rehabilitation lasted between three to ten months [3,4,6,7,10]. A limitation of the few existing studies and case reports is that they use different and not sufficiently described rehabilitation protocols [14]. Additionally, next to standard clinical examinations and imaging procedures, the use of biomechanical tests throughout the rehabilitation after a surgical treatment of a proximal rupture of the rectus femoris muscle were only applied in one case report [15]. The case shows that the isokinetic measured quadriceps strength was restored six months post-surgery; particularly, at joint positions with longer muscle lengths [15]. To the best of our knowledge, one-year follow up data have not been shown so far, which limits the insights into more long-term outcomes.

Overall, there is a lack of research concerning substantial information for the operative treatment and long-term outcome of a complete avulsion of the proximal rectus femoris muscle in terms of the exact injury mechanisms, surgical procedure, and rehabilitation protocol. Therefore, this study aimed to provide a detailed case report addressing all these points over a one-year follow-up period. To obtain more functional insights into the rehabilitation, several biomechanical tests were also conducted.

## 2. Case Presentation

### 2.1. Anthropometrics and Injury Mechanism

The patient was a 41-year-old recreational endurance athlete, who trained in running and cycling three to six times per week over the last 15 years. Sixteen years ago, he had a right medial meniscus tear and anterior cruciate ligament reconstruction using a semitendinosus graft. Concerning the complete avulsion of the proximal rectus femoris muscle, the patient reported the following injury mechanism: During a pass in a leisure soccer match, exceptionally carried out once, the right foot was slightly caught the ground. The patient felt a sharp pain at the right anterior hip region and stopped playing soccer immediately. Walking was uncomfortable and pressure pain was localized proximally at one third of the thigh. No acute swelling or hematoma was observable. An inadequate first on-field treatment was conducted, namely cooling, long-lasting sitting, and drinking 0.8 L beer. Three hours after the injury mechanism, the patient cycled 6 km by bike back home. Between the injury mechanism and the first medical diagnosis two days later, the patient himself used crutches and compression tape to immobilize the leg during daily life.

### 2.2. Diagnosis and Surgery Procedure

The clinical diagnosis was stepwise carried out by clinical examinations as well as ultrasound and magnetic resonance imaging (MRI) procedures two and three days post-injury, respectively. The initial clinical examination showed a palpable defect about 10 cm distal to the origin of the rectus femoris muscle. The patient was unable to voluntary activate the rectus femoris and could not perform an active straight leg raise. The ultrasound imaging showed a large hematoma and an absence of the proximal tendon insertion. To validate, an MRI was conducted showing a grade 4 tendinous avulsion of the rectus femoris with a significant retraction [16]. Clearly visible was a bone-tendon gap and wavy course of the tendon caused by retraction and loss of tension (see Appendix A for anonymous MRI images).

After discussions of the benefits and risks of an operative and a conservative treatment, the patient opted for a surgical repair. Nine days post-injury, the surgical repair of the proximal rectus femoris was performed by using a direct anterior anchor approach to the hip. The initial exposure was carried out through a 10 cm longitudinal incision, which extended distally from the anterior inferior iliac spine. The fascia was opened and the lateral femoral cutaneous nerve was identified and preserved. The preparation followed the interval between the tensor fascia lata and the sartorius muscle. A seroma was drained. Some adhesions were detached, and the retracted proximal tendon was identified and debrided. After further mobilization, the muscle could be approximated to its insertion without relevant tension. An all-suture anchor (JuggerKnot 2.9 mm, Biomet^®^) with two nr. 2 sutures was placed at the anterior inferior iliac spine, and the tendon was secured and anatomically attached using modified Krakow stitches. The patient remained in the hospital for two days. 

### 2.3. Postoperative and Rehabilitation Procedures

During the first post-surgery week, the initial medication consisted of analgesic, opioid, and nonsteroidal anti-inflammatory drugs as well as proton-pump inhibitors and antithrombotic medications. After surgery, the patient reported an irritation of the lateral femoral cutaneous nerve, which was accompanied by a loss of sensation and tenderness on the lateral side of the thigh, which recovered six months post-surgery. For the first six weeks after surgery, the patient used crutches. Weight bearing activities, stretching the quadriceps, and raising the straight leg were prohibited.

From the 4th to 28th day post-surgery, a continuous passive motion device was used two times a day for 90 min to mobilize the knee and hip joint and to prevent adhesions, since active hip flexion was not allowed. The knee and hip flexions were initially restricted and then stepwise increased until the 10th day from 5–60° and 10–45° to 0–90° and 5–80°, respectively. From the 6th to 42nd day, the patient walked 30–60 min with restricted weight bearing each day. On the 9th day post-surgery, a calf muscle venous thrombosis was diagnosed in the operated leg, which was treated with three initial heparin injections (Clexane^®^ 8000 I. E. 80 mg), compression socks, and blood-thinning pills (Elequis 2 × 5 mg per day) for the next three months. Potentially due to the high-dose injections, a hematoma of the size of a tennis ball was observed under the scar approximately 10 cm distally to the origin of the rectus femoris. Manual lymphatic drainage was applied to sustain a reduction of the swelling, which was achieved within the next 14 days. Between the 7th and 13th week post-surgery, physiotherapy sessions were implemented once a week, including strengthening and stretching exercises. In consultation with the surgeon, the patient started self-conducted training on a stationary cycling ergometer on the 36th day post-surgery. The duration and power of all training sessions were monitored and are presented in Figure 1. After getting permission to fully weight bear on the 42nd day post-surgery, the patient walked 4 km through hilly terrain three times per week for the next three weeks. The daily life activities (e.g., climbing stairs) were stepwise increased. On the 59th and 64th day post-surgery, the patient was able to start cautiously with outdoor-cycling and jogging, respectively. Without any complications, the intensity and duration of the cycling and jogging training sessions were stepwise increased during the following 10 weeks until the pre-operative training volume was reached. 138 days (about 20 weeks) post-surgery, the patient was able to finish a 20 km run.

### 2.4. Testing Procedures

By chance, pre-injury values of the endurance performance were available, as the patient was tested eight days before the injury. Furthermore, during several time points of the rehabilitation and one year after surgery, the patient underwent different biomechanical testing procedures under standardized laboratory conditions. The methods and results are briefly introduced in the following sections.

#### 2.4.1. Maximum Voluntary Neuromuscular Activation within the First Nine Weeks Post-Surgery

To investigate the initial healing of the muscle-tendon unit, the ability of the patient to voluntarily activate and contract the rectus femoris muscle was assessed for the first nine weeks after surgery. Between the 18th and 60th day post-surgery, the neuromuscular activation of the rectus femoris muscle was measured weekly during three maximal voluntary unilateral contractions of 5 s at an isometric position of 90° knee and hip flexion. A bipolar surface electromyography system (Noraxon Telemyo DTS, Scottsdale, AZ, USA) was used with a sampling rate of 1500 Hz. The application of the electrodes (Noraxon Dual, Scottsdale, AZ, USA) on the rectus femoris muscle was performed in accordance with the SENIAM (surface electromyography for the non-invasive assessment of muscles) recommendations [17]. The signals were rectified, time-normalized, and smoothed using a root-mean-square moving-window function with a time constant of 200 ms. Figure 2 (top) shows the mean activation of the operated muscle at different time points during rehabilitation compared to that of the non-operated side on the 60th day post-surgery. The mean of the median frequencies of the corresponding raw signals are listed at the bottom of Figure 2.

The data show that a first noticeable neuromuscular activation of the operated rectus femoris muscle was detectable at the 32nd day post-surgery. Thereafter, a weekly increase in the amplitude of the signal was found. At the 60th day post-surgery, the amplitudes of both sides were comparable. In contrast, the median frequency increased initially from the 25th to the 32nd day post-surgery and remained stable thereafter. Even at the 60th day post-surgery, a significant reduction in the median frequency of the operated leg compared to that of the non-operated leg was present.

#### 2.4.2. Crank Torque during Ergometer Cycling within the First 9 Weeks Post-Surgery

The patient frequently trained on a stationary cycling ergometer (Lode Excalibur Sport, Groningen, The Netherlands) equipped with crank torque sensors from the 36th to the 58th day post-surgery. The duration and power of all training sessions are shown in Figure 1. During all sessions, the bilateral crank torque was routinely measured at 2° intervals to document potential changes in leg symmetry. In summary, more than 71,400 revolutions were performed and analyzed. For each time point and power stage, a mean crank torque curve was calculated for the operated and non-operated side. The difference between both legs was calculated and related to the total average over one revolution to account for the different power stages. The mean difference curves are plotted in Figure 1.

During rehabilitation, the intensity and volume of the training sessions were stepwise increased in a self-driven manner up to 200 W on the 58th day post-surgery. With the exception of the first training session, all difference curves had a similar progression with the highest positive and negative differences between a crank angle of 60–90° and 270–310°, respectively. These positions correspond to the knee extension and hip flexion phases of the rectus femoris muscle. The presented data show that the side differences in crank torque between the non-operated and operated leg considerably decrease during the rehabilitation.

#### 2.4.3. Neuromuscular Activation during Ergometer Cycling on the 60th Day Post-Surgery

At the end of the ergometer training period, the bilateral symmetry of the neuromuscular activation of the rectus femoris muscle was additionally evaluated. Surface electromyography (SEMG) was measured on the 60th day post-surgery during the ergometer cycling at different intensities. The preparation and processing of the electromyographic measurement was the same as mentioned before with the exception of a smaller root mean square (100 ms). The patient cycled at 50, 100, and 200 W with a constant cadence of 80 rpm. Figure 3 shows the mean neuromuscular activation of the operated and non-operated rectus femoris muscles. 

The data demonstrate noticeable side differences in the neuromuscular activation at 50 and 100 W during both the hip and knee flexion phase. However, at 200 W, the side differences decrease and a prominent difference persists only at the beginning of the hip flexion phase (upwards movement of the leg). It is noteworthy that with increasing power, the neuromuscular activation of the non-operated rectus femoris decreases during the knee extension phase.

#### 2.4.4. Leg Volume during the One-Year Follow-Up Period

To quantify potential atrophic and hypertrophic effects, the volume of both legs was assessed at different time points throughout rehabilitation using the Perometer 350 T (Pero-System GmbH, Wuppertal, Germany). The device measures the maximum horizontal and vertical diameters at intervals of 4.7 mm along the longitudinal axis of the limb and calculates segmented and total volumes. Figure 4 shows the total leg volume of the operated and non-operated leg at nine different time points as well as the segmented volumes and side differences on the 18th, 60th, and 186th day as well as one year post-surgery.

The measurements show that the total leg volume of the non-operated leg increased stepwise from the 18th to the 53rd day post-surgery (see top of Figure 4). Additionally, the volume of the operated leg increased during rehabilitation. However, this increase seems to be delayed, and therefore the maximum absolute difference between both legs was detected on the 46th and 53rd day post-surgery. About six months (186 days) post-surgery, the total volume was comparable between both legs. The segmented volumes show that the greatest changes in both legs occurred at the middle calf and middle thigh (see Figure 4, middle). The progression of the side differences in the segmented leg volume were comparable between the 18th and 60th day post-surgery but differ compared to those at six months post-surgery. The calf volume of the operated leg was considerably higher than that of the non-operated leg, while the side difference in the thigh volume was the lowest. While the total volume was comparable between both legs at six months post-surgery, substantial side differences in the segmented volume persist. Even at one year post-surgery, differences in the calf and thigh volumes of both legs were present.

#### 2.4.5. Endurance Performance during the One-Year Follow-Up

To gain additional insights into the recovery of the endurance performance, incremental exercise tests with respiratory gas exchange measurements on a cycling ergometer (Lode Excalibur Sport, Groningen, The Netherlands) were conducted 6.5 and 9.0 months post-surgery. The data were related to those assessed, 17 days prior to the injury. Each incremental test started at 100 W and increased every 3 min by 30 W until exhaustion was reached. During the test, respiratory gas exchange and heart rate were measured using an open-circuit breath-by-breath gas analyzer (Ganshorn, PowerCube-Ergo, Niederlauer, Germany) and short-range telemetry (Polar, T31, Kempele, Finland). The peak oxygen uptake and heart rate were defined as the highest recorded data during the test. Peak power was calculated by linear interpolation according to the last fulfilled power stage.

The heart rate, oxygen, and carbon dioxide uptake values of the three tests were plotted in Figure 5. The data show that the peak power output increased from the pre-injury value of 352 W to 368 W and 400 W after 6.5 and 9.0 months post-surgery, respectively. The corresponding peak oxygen uptake and heart rate values were 58.7, 54.1, and 55.9 mL/min/kg and 172, 177, and 178 bpm, respectively. Overall, the aerobic endurance performance at 6.5 months post-surgery was comparable to the pre-injury level. However, at 9.0 months post-surgery, the power output, as the most crucial key factor of endurance performance, was highest.

#### 2.4.6. Anatomical Quadriceps Characteristics at One Year Post-Surgery

To evaluate the anatomical characteristics of the quadriceps muscles, an MRI using a 3T scanner (Magnetom Vida; Siemens Healthcare, Erlangen, Germany) with an 18-channel anterior body coil in combination with a 32-channel spine coil of both thighs was performed at one year post-surgery. A transversal T1 weighted turbo spin echo Dixon sequence and a transversal 3D T1 weighted vibe sequence were acquired (slice thickness 1.2 mm). From the Dixon sequence, water-only, fat-only, in-phase (water and fat), and out-of-phase (water minus fat) images were created. Signal intensity values of in-phase images and fat-only images for the rectus femoris, vastus medialis, vastus lateralis, and vastus intermedius muscle were measured for quantitative analysis of intramuscular fat content. A region of interest neatly encircling the borders of each muscle were placed on every slice of the in-phase images and copied over to the fat-only images. Fat fraction was calculated from the signal fat-only divided by the signal in-phase [18]. From the vibe sequence, consecutive images were obtained for the manual 3D-segmentation of the entire quadriceps muscles with a post processing computer software (3D-Slicer 4.10) [19]. After applying a Gaussian smoothing method (3 mm), muscle volumes were quantified using the pixel count method. Additionally, the volume of the femur and part of the pelvis was bilaterally assessed from the same MRI sequence to estimate the precision of this procedure.

As shown in Table 1, the quadriceps muscle volumes of the operated leg were 10–27% lower than those of the non-operated leg. In contrast, the bone volumes (≤3%) and fat fraction of the quadriceps muscles (≤4%) did not significantly differ between both legs.

#### 2.4.7. Neuromuscular Activation and Isometric Strength at One Year Post-Surgery

To evaluate the neuromuscular characteristics of the rectus femoris muscle at one year post-surgery, isometric knee and hip strength tests were performed. After a 15 min warming-up procedure on a cycling ergometer, the patient performed unilateral isometric strength tests in eight different positions (see Figure 6) in a standard device (Cybex NORM, Humac, CA, USA). All tests started with the non-operated leg. Joint torques and neuromuscular activation of the quadriceps muscles were assessed at each position during three trials over 10 s in duration. Gravity corrected peak torques and maximum neuromuscular activation were calculated for each position. The preparation and processing of the electromyographical measurement was the same as mentioned before.

The data show that all peak isometric joint torques were lower for the operated leg (−3 to −35%) compared to the non-operated leg with the exception of both hip flexion torques at 60° hip flexion (hip position 1: +7%; hip position 2: +12%). Additionally, the peak neuromuscular activation of all quadriceps muscles was lower in the operated leg (−12 to −50%), with the exception of the rectus femoris muscle at two positions (knee position 1: +14%; hip position 2: +45%).

Overall, side differences in knee extension torque and neuromuscular activation of the rectus femoris muscle increase with greater moment arms at the hip joint and longer lengths of the rectus femoris muscle (position 1 vs. 3 and position 2 vs. 4). During the hip flexion tests, significantly lower values of torque and neuromuscular activation at the operated side were present at 0° hip flexion (position 3 and 4) compared to the non-operated side.

#### 2.4.8. Crank Torque and Neuromuscular Activation during Ergometer Cycling at One Year Post-Surgery

One year post-surgery, the functional characteristics of the quadriceps muscles was evaluated via ergometer cycling with different power and cadence. The bilateral symmetry of the crank torque and neuromuscular activation of the quadriceps femoris muscles during ergometer cycling were synchronously measured at six 2 min stages. Equipment, signal measurement, and processing were in line with those mentioned for the ergometer cycling before. The mean torque and neuromuscular activation graphs were than calculated and are displayed in Figure 7.

Our data display that the side comparison of the neuromuscular activation of the vastus medialis and vastus lateralis muscles revealed different shapes between the operated and non-operated leg; especially during the knee extension phase (0–60° crank angle). Moreover, the neuromuscular activation of the rectus femoris muscle of the operated leg is lower than that of the non-operated leg in the knee extension and the hip flexion phase. However, crank torque graphs of both legs show comparable shapes, whereby the amplitudes of the knee extension phase were slightly lower in the operated leg. Overall, the resulting torque of consecutive crank cycles seems to be constant and symmetrical.

## 3. Discussion

### 3.1. Injury Mechanism, Diagnosis, and Surgery Procedure

In our case report, the injury mechanism equates to one of the most commonly described injury situations in soccer [6,20]. Without the involvement of an opposing player, the kicking leg caught the ground, which leads to a sudden resistance against a forceful concentric muscle contraction [7]. This high impact change from concentric to isometric/eccentric muscle load may have overreached the maximal tolerable strain of the proximal rectus femoris muscle tendon. Within our case report, the impact of the inadequate initial treatment is unknown, but the importance of the initial therapy in muscle injuries has to be acknowledged [21]. The bearable acute pain and absence of a fall were in contrast to the severity of the injury and international muscle injury classification [16]. Thus, a fast and proper diagnosis is of crucial importance for optimal treatment. As shown in Figure 8, a safe and reliable diagnosis of a complete avulsion of the proximal rectus femoris muscle based on ultrasound images alone is difficult. As the tendon is less supplied with blood, only a small hematoma is often detectable and the severity of the injury may be underestimated [6]. However, a complete function loss, palpation of a gap, and/or retraction of the muscle during the physical examination has to be taken seriously [16]. Thereon, and as supported by our data, a fast evaluation per MRI is clearly recommend, which permits a precise localization and quantification of the injury [10,22,23]. In case of a complete avulsion of the proximal rectus femoris with a dislocation and significant retraction, a surgical treatment is recommended [24], because the effect of non-anatomical healing on muscle strength and function is currently unknown [20]. However, among other aspects, the decision has to take into account the age, previous injuries, comorbidities, and activity level of the patient. In particular, in elite athletes, a non-operative treatment may predispose the patient to a higher recurrence rate [10]. As the patient of this case study was a healthy, middle-aged, and highly active athlete, a surgical repair with a suture anchor approach was performed.

### 3.2. Rehabilitation and Testing Procedures until Six Month after Surgery

In general, healing of repaired tendons follows the typical wound healing course, including an early inflammatory phase, followed by proliferative and remodeling phases [25]. Within our study, the first noticeable neuromuscular activation of the rectus femoris muscle was detectable within the 5th week post-surgery. As the muscle could not be voluntarily activated during the first month post- surgery (Figure 2), it is questionable to perform training sessions with the aim to activate selected muscles. Whether the execution of various movements within those training sessions results in enhanced inter-muscular compensation strategies requires further study. A potential reason for the observed neuromuscular function could be a muscular inhibition based on the structural damage of the tendon. As the active muscle structure and the innervation path of the muscles are likely intact, inhibition may be triggered by the acute inflammation and/or pain [26]. Thus, the use of surface electromyography during rehabilitation may be helpful to evaluate the ability for the voluntary neuromuscular activation of the injured muscle. Beside those active muscle contractions, the role of continuous passive movements is unclear, but may also lead to a small mechanical load, which can improve the tendon healing during the proliferation phase [27]. A counterbalance between passive and active tendon load might result in optimal tendon healing [28]. Whether the application of electrical muscle stimulation and/or platelet rich plasma injection may facilitate the voluntary muscle activation within the rehabilitation is currently unknown [24,29]. Since no voluntary contraction was observable in the first month in our report, active muscle training of the injured muscle seems to be impossible. Thereafter, a cautious increase in voluntary muscle activation may be beneficial. At the 9th week post-surgery, the amplitude of the observed neuromuscular activation was comparable between the operated and non-operated side. In contrast, the median frequency initial increased in the 5th week post-surgery and no further change was observed (Figure 2). Compared to the non-operated side, the median frequency was still decreased nine weeks post-surgery. The amount of this decrease was also reported after surgical reconstruction of the anterior cruciate ligament three months post-surgery and is discussed as a reduced neural activation and a shift in type II muscle fibers [30,31]. It can be assumed that neuromuscular characteristics are still altered after 60 days post-surgery [32].

Within the rehabilitation, instrumented ergometer cycling was frequently performed between the 4th and 9th week post-surgery. With increasing numbers of training sessions, the crank torque differences between the legs were reduced; particularly, at angles of 60–130° and 250–320°, which represent the main knee extension and hip flexion phase, respectively (Figure 1). The main power output was initially produced with the non-operated leg, showing the presence of an inter-limb compensation strategy. In cycling, an inter-limb strategy has been shown for other injuries and selected time points [33,34], but not its successive reduction during the rehabilitation as in this case report. Beside the reduction of the inter-limb compensation strategy, the power output and the training duration was stepwise increased. Notably, after three weeks of frequent stationary cycling, the patient was able to get back on his bike and train on the road.

In our case report, we have presented the neuromuscular activation during cycling at different intensities at the 9th week post-surgery (Figure 3). A normalization of the neuromuscular activation could be detected at higher power values; especially during the knee extension phase. However, to the best of our knowledge, this aspect has never been shown for rehabilitation procedures of muscle-tendon injuries before. At this rehabilitation phase, side differences in the neuromuscular characteristics of the rectus femoris muscle were still present and have to be addressed by training exercises with respect to the structural conditions. Deduced from our results, short training sessions with higher loads seems to be a promising approach to normalize muscle function, but this must be confirmed by further studies. 

During rehabilitation, the total leg volume increased in both legs, while the increase in the operated leg occurred with a time delay (Figure 4). However, six months post-surgery, the side differences in the total leg volume was lowest. The segmented volume graphs clearly showed that the reduction in the side differences were based on a considerable increase in the calf volume of the operated leg. Thus, a muscular intra-limb compensation strategy may have led to an adaptation of the calf muscles. The use of those intra-limb compensation strategies was also described after knee injuries [35,36], which often result in a long-term muscle weakness of the quadriceps [37]. However, whether those asymmetries and compensation strategies may be helpful to reduce the load of the altered resilience of the anatomical structure (e.g., muscle-tendon unit) or, in the long-term, lead to serious complications must be discussed for each patient on an individual basis [38]. Despite the remodeling of an injured tendon, the biochemical and mechanical properties never match those of an intact tendon [28]. From our perspective, during the rehabilitation after surgery, an unconscious self-organization of the sensorimotor system occurs, and patients have to use their degrees of freedom to fulfil the challenges in daily life and/or sport [39]. Thereby, they have to consider their individual requirements (e.g., movement experience, anatomic structure, proprioception, etc.). Finally, some patients indeed return to pre-injury level with normal or adapted function, while some never reached their prior level [40]. Notable is that the cycling endurance performance of our patient was comparable to pre-injury levels at six months after surgery (Figure 5), while neuromuscular muscle activation and segment volumes showed significant side differences.

### 3.3. One-Year Follow up Testing Procedures

Another aim of this case report was to evaluate the long term outcome (e.g., structural and functional characteristics of the rectus femoris muscle) at one year after surgery. To the best of our knowledge, no study is available showing any biomechanical long-term evaluation after a complete avulsion of the proximal rectus femoris muscle. Generally, the patient reported no limitations during his jogging and cycling training as well as activities of daily life.

However, the aforementioned intra-limb compensation, which was shown by the leg volume at six months after surgery, was already present one-year after surgery (see Figure 4), at which only the thigh volume in the non-operated was further increased. The long term adaption in the thigh volumes may reflect that the motor control strategy, which was present at six months persists permanently. This is also derivable by the side differences in the muscle volumes of the quadriceps, which were revealed by the MRI, but not by differences in the fat content of the muscle tissue (Table 1). The overall side differences of the quadriceps muscles volume in our study was 18%, which is comparable to that of patients with hip osteoarthritis (20%) [41]. From this perspective, an (long-term) adaption in muscle volumes may be useful to detect compensation strategies and consequently a change in motor control.

Additionally, the side differences in knee extension and hip flexion strength as well as in the neuromuscular activation of the quadriceps muscles reveal long-term adaptions based on changed motor control strategies (Figure 5). Interestingly, side differences of the rectus femoris muscles during knee extension were more pronounced at longer muscle lengths. Lower neuromuscular activation in the operated rectus femoris muscle and higher torque values were found for hip flexion at a hip flexion angle of 0°. It is presently unclear whether those side differences can be reduced by (angle) specific (strength) training and whether that makes sense regarding potential asymmetric characteristics of the muscle–tendon system. It remains unclear whether the original geometric position of the rectus femoris was completely restored. A small change in tendon length and/or position of the origin may lead to asymmetric moment arms and/or muscle lengths at the same joint positions. This aspect further requires a changed motor control during uni- and bilateral movements even after a non-operative treatment [13]. Therefore, this case report underlines that more individualized assessment tools and rehabilitation procedures are required.

As the patient performed cycling as his main hobby, we analyzed this bilateral cyclic movement at different power and cadence to assess functional characteristics of the quadriceps muscles and crank torque side differences. Even during cycling, side differences were present in the neuromuscular activation of the quadriceps muscles (Figure 6). The SEMG activation of the rectus femoris muscle of the operated leg was lower than that of the non-operated leg. Moreover, the activation pattern of the vastus medialis and lateralis muscles also showed asymmetric shapes. However, despite the asymmetric muscle activation, the resulting crank torque (movement outcome) was almost symmetric. Therefore, the adapted motor control strategy during cycling may biomechanically be effective as a change in the hip flexion phase of one leg has an influence on the knee extension phase of the contralateral side.

## 4. Conclusions

Within our case report, we have presented the surgical treatment of a complete avulsion of the proximal rectus femoris muscle including monitored rehabilitation phases and comprehensive biomechanical test procedures. Given the rarity of this severe injury, much information can be expected for the treatment and future research:Within the first month post-surgery, an almost total neuromuscular inhibition of the rectus femoris muscle was present.A stepwise reduction in inter-limb compensation was observable (e.g., in crank torque) during rehabilitation.Muscular intra-limb compensations were shown at six months post-surgery and even one year after surgery, which were also represented in the long-term adaption of the muscle characteristics and leg volumes.A changed motor control strategy can result in asymmetric muscle patterns during ergometer cycling, and can even result in symmetric power output.During rehabilitation, there might be a benefit to normalizing neuromuscular muscle activation in ergometer cycling using higher loads.

## 5. Additional Comments

While writing the manuscript, the patient reported a partial muscle tear (type 3b) in the middle part of the operated rectus femoris muscle during a daily life activity [16], which was about 3.5 month after the one-year follow-up.

## Figures and Tables

**Figure 1 ijerph-18-08727-f001:**
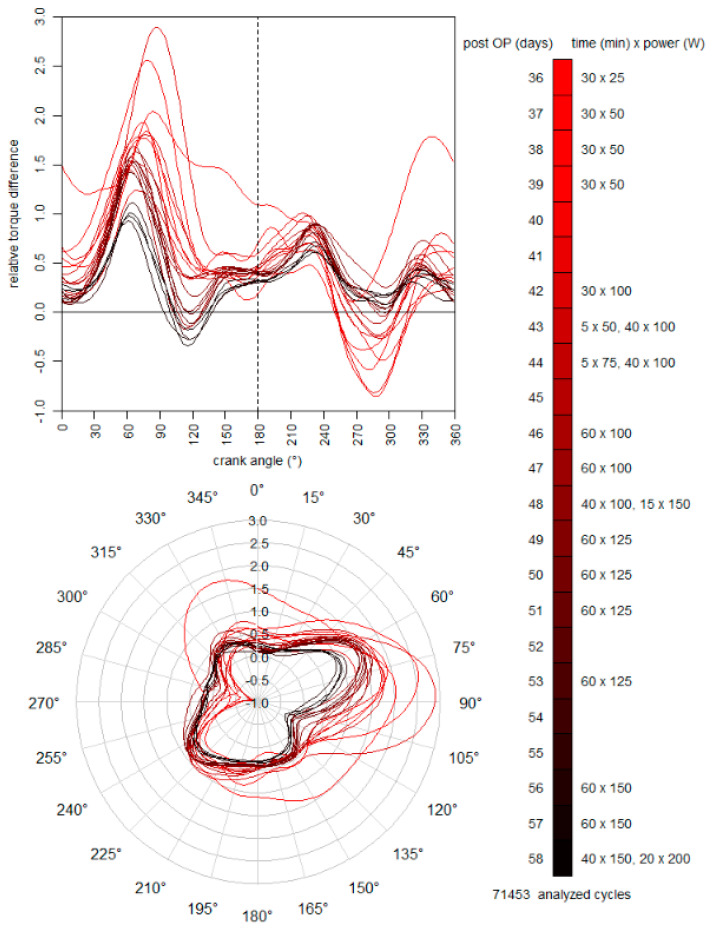
Torque difference between the operated and non-operated leg relative to the mean overall torque during ergometer cycling training sessions of different duration and power from 36 to 58 days post-surgery.

**Figure 2 ijerph-18-08727-f002:**
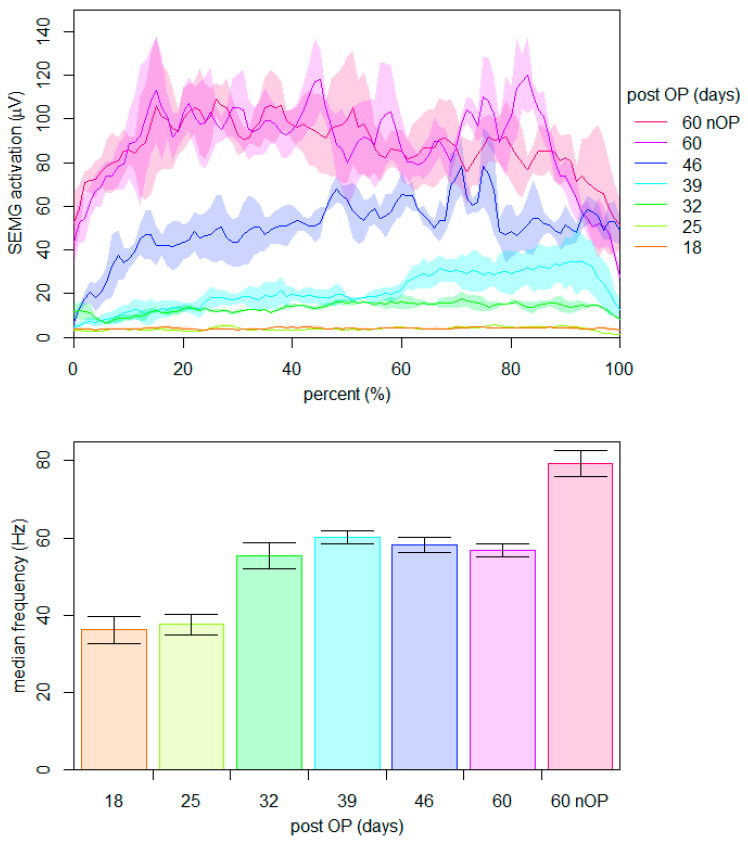
Surface electromyographic measurements of the operated rectus femoris muscle at six different time points during rehabilitation compared to the non-operated side. Note: mean activation and sd of three maximum voluntary time-normalized contractions (**top**) and mean and sd of the corresponding median frequencies (**bottom**).

**Figure 3 ijerph-18-08727-f003:**
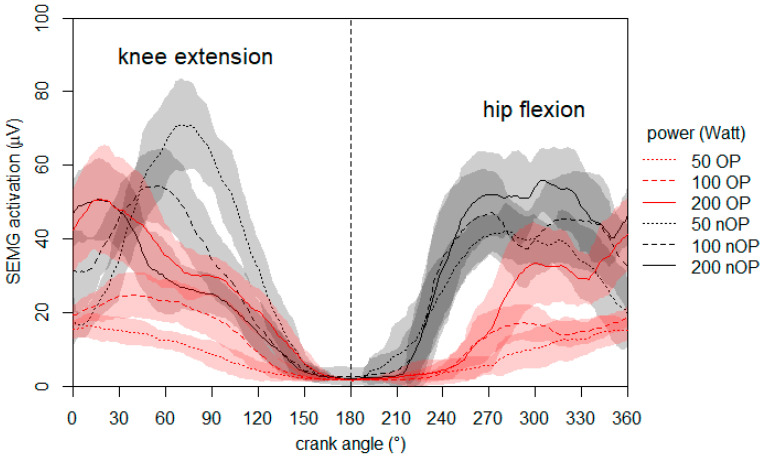
Mean neuromuscular activation (SEMG) of the operated (OP) and non-operated (nOP) rectus femoris muscle during cycling at 50, 100, and 200 Watt measured 60 days post-surgery.

**Figure 4 ijerph-18-08727-f004:**
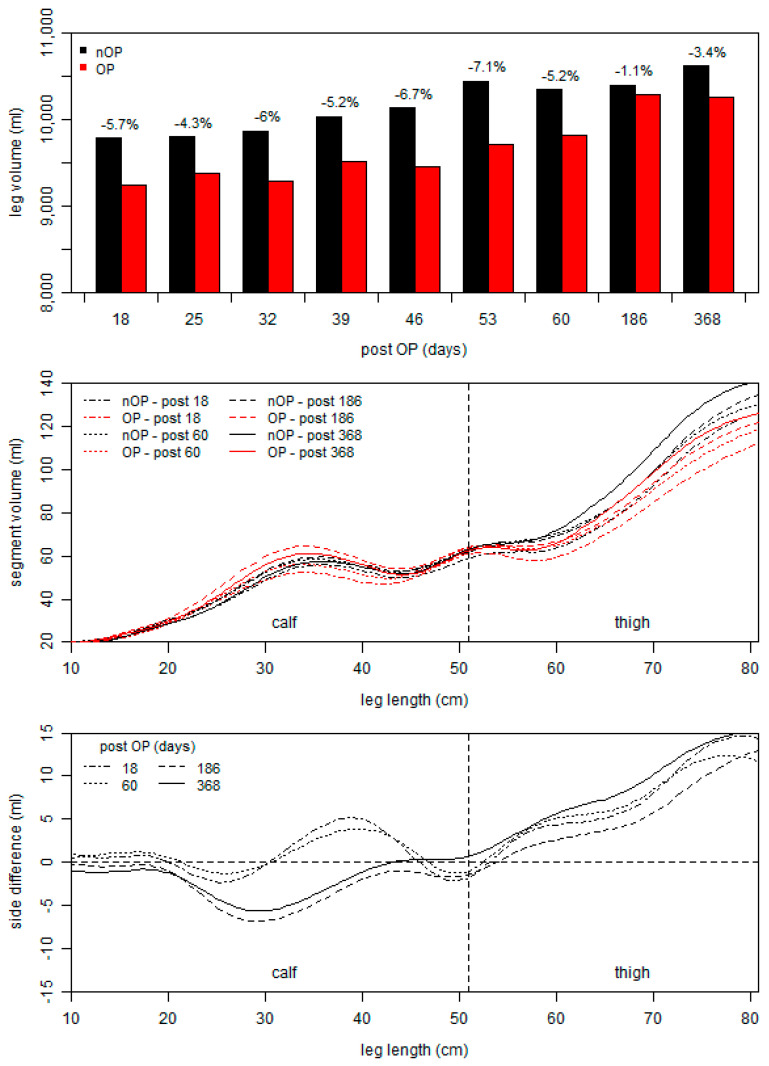
Total leg volumes of the operated (OP) and non-operated (nOP) leg (**top**), segment volumes (**middle**) and side differences (**bottom**) at the 18th, 60th, 186th, and 368th day post-surgery.

**Figure 5 ijerph-18-08727-f005:**
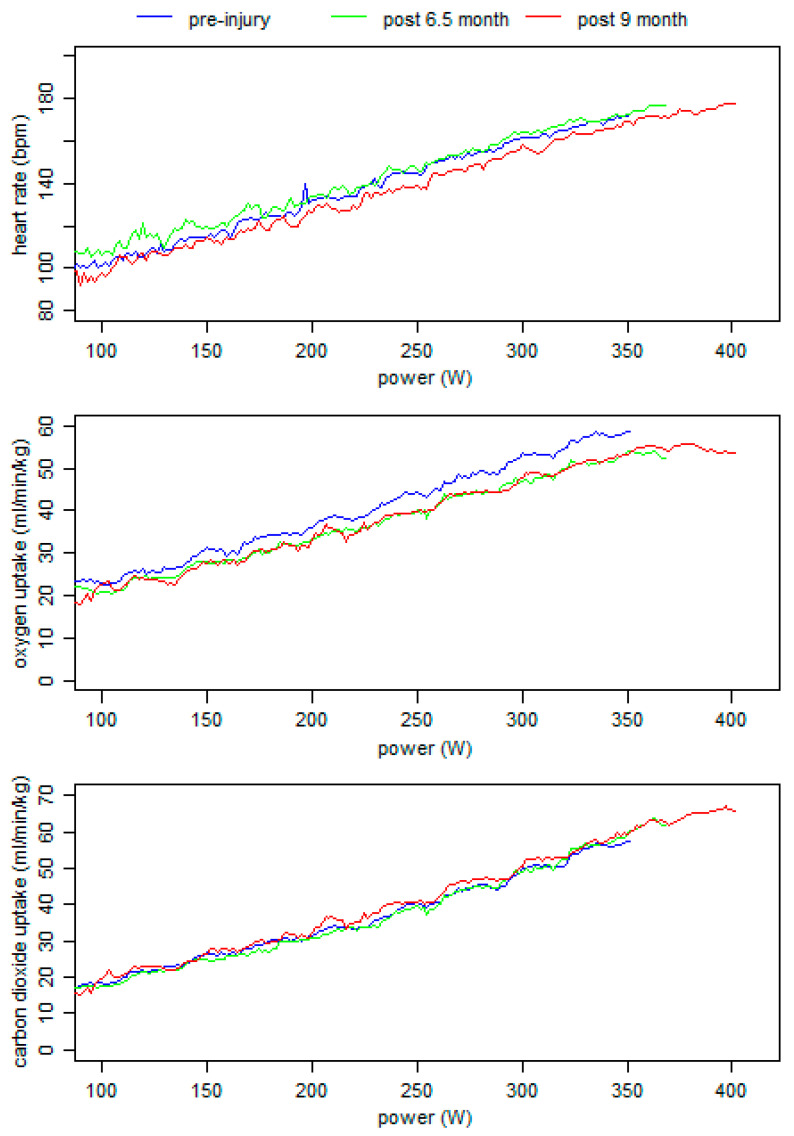
Heart rate, oxygen, and carbon dioxide uptake values during the three incremental exercise tests.

**Figure 6 ijerph-18-08727-f006:**
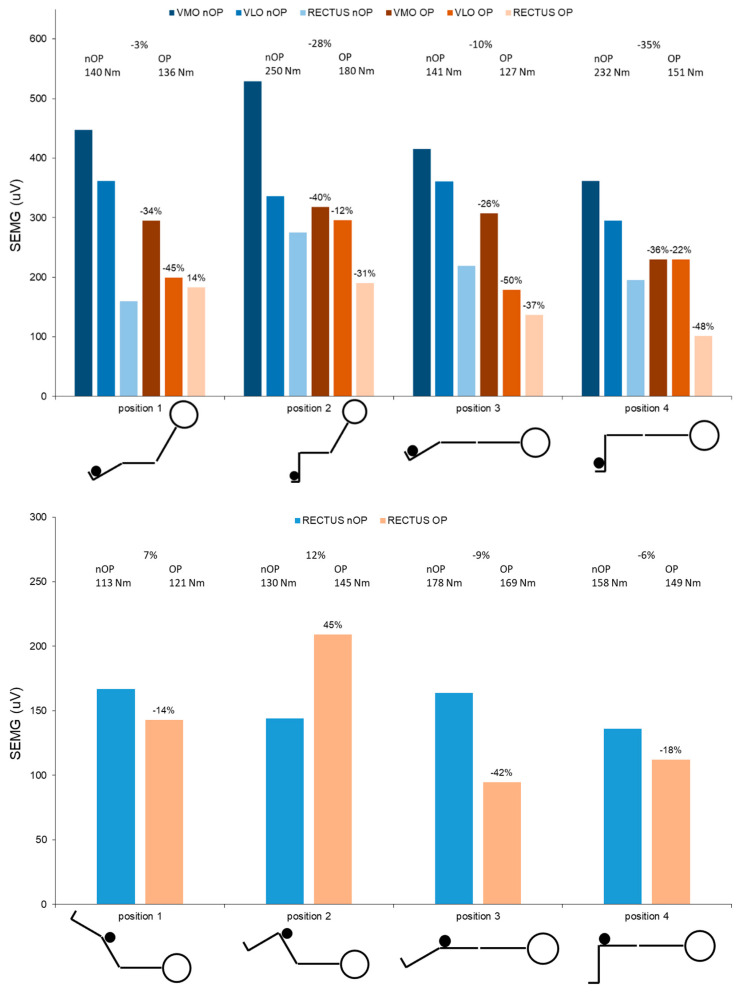
Peak neuromuscular activation (SEMG) during isometric knee extension (**top**) and hip flexion (**bottom**) of the quadriceps muscles and corresponding peak torque values at different joint positions separated for the operated (OP) and non-operated (nOP) side at one-year post-surgery. (VMO—m. vastus med., VLO—m. vastus lat., RECTUS—m. rectus fem., %—percentage side differences).

**Figure 7 ijerph-18-08727-f007:**
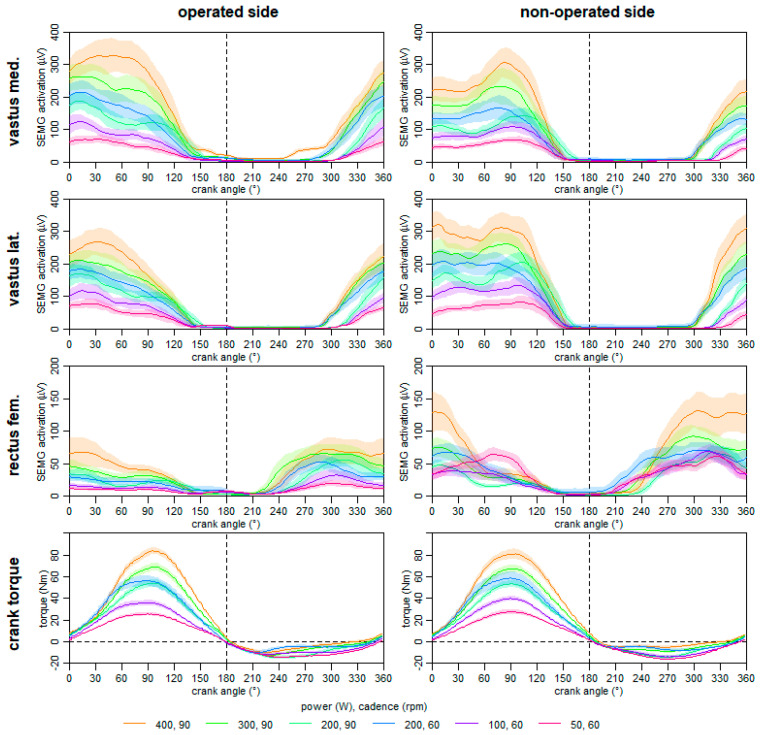
Mean neuromuscular activation (SEMG) of the quadriceps muscles and crank torque graphs during ergometer cycling at different power and cadence separated for the operated and non-operated side at one year post-surgery.

**Figure 8 ijerph-18-08727-f008:**
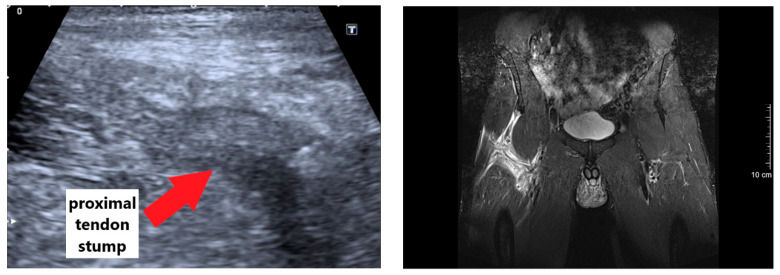
Axial and coronal magnetic-resonance (**right**), axial and sagittal ultrasound (**top left**), and intraoperative (**bottom left**) images.

**Table 1 ijerph-18-08727-t001:** Fat fraction and muscle volume of the quadriceps muscles, pelvis, and femur revealed from the MRI one-year post-surgery.

	Side	M. Rectus Femoris	M. Vastus Intermedius	M. Vastus Lateralis	M. Vastus Medialis	Femur	Pelvis
Fat fraction	nOP	0.105	0.112	0.109	0.112	-	-
	OP	0.106	0.109	0.105	0.114	-	-
	OP/nOP	−1%	3%	4%	−2%		
Muscle volume (cm³)	nOP	355.2	897.9	1033.0	730.8	590.6	303.5
	OP	287.3	654.0	931.5	617.4	586.0	295.4
	OP/nOP	19%	27%	10%	16%	1%	3%

## Data Availability

The datasets used and/or analyzed during the current study are available from the corresponding author upon reasonable request.

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
