# Peer review of "Rehabilitation after a Complete Avulsion of the Proximal Rectus Femoris Muscle: Considerations from a Case Report"

_ijerph, 2021, doi:10.3390/ijerph18168727_

Round 1

Reviewer 1 Report

A very long article for one case report. Methods and results are described in too much detail.
It is likely that the article may strengthen the comparison with other type of treatments. For example - Esser S et al. Proximal Rectus Femoris Avulsion: Ultrasonic Diagnosis and Nonoperative Management.

Author Response

We would like to thank the reviewer for his positive feedback. We agree that our article is long; however, it should be considered that we have applied a comprehensive biomechanical approach, including new and previously not described methods, to increase the understanding of the rehabilitation after a complete avulsion of the proximal rectus femoris muscle. Nevertheless, in our revised version, we have carefully reduced the number of words and hope that the reviewer can accepted our proceed. Moreover, in the discussion, we have considered the suggested article again. Thank you for your time and help to increase the quality of our paper.

Reviewer 2 Report

Thank you for your interesting and extensive case report.

An extensive review of the writing is needed, both grammar (e.g. under 1. Background (…)one-year follow up data have not been shown never shown before(…) and vocabulary. Alot of connective words have been used in this text (i.e. moreover, therein, etc.), these should be reduced by a significant amount to improve readability.

While the postoperative procedure is described well, it is unclear how the weight bearing was restricted/controlled.

I appreciate the great detail in which examinations of the fuction of the muscle have been performed. A comparison to a healthy individual or data from an individual suffering from a different injury of the thigh might be helpful. It is mentioned that pre-injury endurance testing was performed by chance, these results should be included in the graphs.

Limitations of the methods should be included in the discussion, especially considering the Patient had previously had a significant injury, muscle harvesting and tendon graft to acl reconstruction with subsequent immobilization to the same leg.

Author Response

We would like to thank also this reviewer for his interest, time, and effort to review our manuscript. Based on your constructive feedback, we have revised our paper and now submit an improved version. We have carefully revised the language and grammar, and reduced much of the connective words. Moreover, we have clarified how the weight bearing was controlled. The fact that the patient had a previous injury to the same leg 16 years ago is already addressed in the case presentation. I’m sorry, but we disagree that this is a limitation within a case study. Concerning the pre-injury endurance data, we agree that it is rational to display the endurance data in a further figure, which we added to the paper. Since every sport injury and patient is unique, we did not compare our data with those from a different injury or another patient etc. A comparison is also challenging due to the fact that we were the first to apply comprehensive biomechanical testing procedures additional to standard medical examinations. We hope that you can understand our proceed and thank you again for your help to improve the quality of our manuscript.